# Understanding the Changes in Social Support Expressed in a COVID-19 Online Forum

## Abstract

In online forums focused on health-care and well-being, individuals tend to seek and give the following social support: emotional and informational support. Understanding the expressions of these social supports over time in an online COVID-19 forum is important for (a) the forum and its members to provide the right type of support to individuals seeking support and (b) determining the long term effects of the pandemic on the well-being of the public, thereby informing interventions. In this work, we build four machine learning (ML) models to measure the extent of the following social supports expressed in each post: (a) emotional support given (b) emotional support sought (c) informational support given, and (d) informational support sought. For each post, these ML models output a numerical value indicating the extent of each of these social supports expressed in the post. We aggregate these social supports expressed in posts by averaging their scores on a weekly basis and determine how these social support sought and given changes over time in posts published in an online COVID-19 forum.

## 1 Introduction

Globally, millions of individuals have contracted COVID-19 and more than 600,000 have died from the pandemic as of July 2020. Individuals are turning to online forums focused on discussions around COVID-19 to seek and give support and posts from these forums can be used to determine patterns in public discussions around COVID-19 and how these patterns change over time (Stokes et al., 2020). In online forums focused on health and well-being, individuals tend to seek and give two forms of social support: emotional support and informational support (Wang et al., 2012; Yang et al., 2017); examples (paraphrased) of posts that express these social supports are:

**Social Support examples**

- **Emotional Support Given:** *"I wish everyone on this forum good health however if some of us end up getting the virus please keep others posted. We will offer emotional support while learning how does it feel to go through it at the same time"*

- **Emotional Support Sought:** *"What is going to happen if we are forced to stay home and not work for a couple of weeks? I can't afford 2 weeks with no pay :( - "*

- **Informational Support Given:** *"A 14-day quarantine makes sense to make sure you are not going to become symptomatic and prevent pre-symptomatic spread."*

- **Informational Support Sought:** *"What happens when we find a vaccine? Do countries pay the lab for a license to produce it internally? "*

Some of these posts may express multiple social supports; the example post above for emotional support sought seeks both emotional support and informational support.

Understanding how these social support needs change over time in a COVID-19 online forum is important for several reasons; first, it can help the forum moderators and members provide adequate and meaningful support to users seeking support, and secondly, it can provide insights as to the long term effects of the pandemic on the well-being of the public, thereby providing the necessary authorities with sufficient and timely information about the changes in social support needs of the public so that they can make resources available to address these needs.

| Emo G | Emo S | Info G | Info S | Post |
|-------|-------|--------|--------|------|
| 1 | 1 | 1 | 5.33 | Does anyone know if you recover from the virus can you get it again? |
| 3.33 | 1 | 1 | 1 | It's absolutely NOT the end of the world. Should we be worried? Yes. |

Table 1: Examples of posts with average ratings for emotional support given (Emo G), emotional support sought (Emo S), informational support given (Info G), and informational support sought (Info S)

In this work, given posts in an online COVID-19 forum, we build 4 machine learning models to measure (i) the extent of emotional support given, (ii) the extent of emotional support sought, (iii) the extent of informational support given, and (iv) the extent of informational support sought, with the aim to determine gaps in social support supply and demand in posts published on an online COVID-19 forum over time.

## 2 Dataset

Reddit is an online forum that is made up of several sub-forums called "subreddits" - each of which is focused on discussions around a particular topic. Our dataset consists of posts published in the /r/Coronavirus subreddit, which focuses on discussions around COVID-19 and is the COVID-19 subreddit with the most number of members (i.e. 2.2 million members as of July 2020). Specifically, we collected 64,074 posts published daily in the "Daily Discussion Post" thread in the /r/Coronavirus subreddit between March 3 and April 30 2020 (Stokes et al., 2020). In our analysis, we exclude the comments these posts received.

## 3 Social Support

The importance of social support expressed in online health forums has been demonstrated (Wang et al., 2012; Yang et al., 2017).

Similar to prior work Wang et al. (2012), we measure the extent of social support expressed in posts; we had 3 medical students rate a sample of 1,000 posts from our dataset. Given a post, the annotators rated the posts on: (i) the extent of emotional support given (ii) the extent of emotional support sought (iii) the extent of informational support given (iv) the extent of informational support sought; where similar to Wang et al. (2012), we define emotional support posts as seeking encouragement, sympathy, affirmation, understanding, or caring; and informational support posts either give or seek advice or information. The annotators rated each post using a 7-point Likert scale, where 1 meant "a particular social support was not expressed" and 7 meant "a particular social support was expressed a lot". For each post, the annotators ratings were aggregated by averaging their ratings, as shown in Table 1. Using intra-class correlation (ICC) (Bartko, 1966), which is commonly used to measure annotator reliability when each post is rated by different groups of annotators, we measured the reliability of the annotators; the ICC for emotional support given, emotional support sought, informational support given, and informational support sought were 0.650, 0.660, 0.778, 0.918, respectively.

Similar to prior work (Wang et al., 2012), we extracted the following features from each post:

### 3.1 Features

Using Linguistic Inquiry and Word Count (LIWC) (Pennebaker et al., 2015) - a dictionary of psycho-linguistic categories such as parts-of-speech and health, we selected LIWC categories which are relevant to emotional and informational support (Wang et al., 2012).

We selected the following language features (Wang et al., 2012): (a) the number of words (b) the number of sentences (c) the number of sentences in a post that mention negation words/phrases such as "couldn't", (d) the number of sentences posed as questions.

Given a post, we counted the number of specific parts-of-speech tags and the number of strong and weak subjectivity words (Wilson et al., 2005) in each post. We identified posts involving requests or advice (Wang et al., 2012).

Using Latent Dirichlet Allocation (LDA) (Blei et al., 2003), we generated 20 topics (we varied the number of topics and 20 topics gave optimal results) from the posts from our dataset. A healthcare professional with a graduate degree manually assigned labels to each of the topics (Wang et al., 2012).

### 3.2 Social Support Prediction Model

Given the language features extracted from posts in section 3.1, we built four machine learning Random Forest models. For each post, these models produced numerical values indicating the extent of (i) emotional support given, (ii) emotional support sought, (iii) informational support given, and (iv) informational support sought. The 1,000

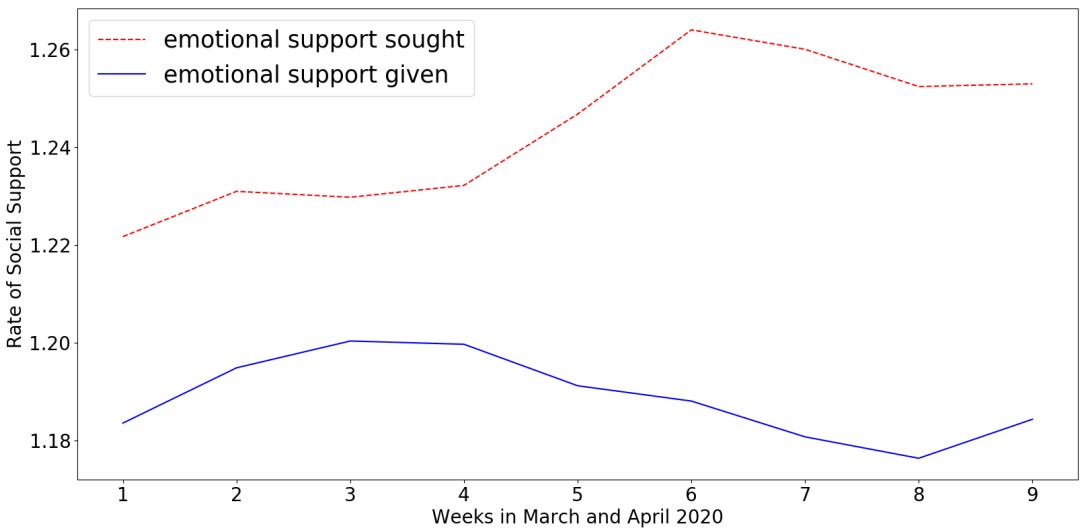

Figure 1: Graph showing the change in the average weekly emotional support sought and emotional support given in posts from our /r/Coronavirus dataset

annotated posts were randomly partitioned and 80% was used for the training set, 10% was used for the validation set, and 10% was used for the test set. We used the validation set to evaluate the performance of the models and when the models performance on the validation set was satisfactory, we then evaluated the model on the test set. Similar to Wang et al. (2012), Pearson's correlation was used to measure the performance of the models. The results from the models were correlated with the average annotator ratings for each post.

**Result from Social Support Prediction Model:** On the annotated dataset (section 3), these models correlated with the average annotator ratings as shown in Table 2. We then applied these models to all the posts in our dataset. Each post ended up having scores for the emotional support given, emotional support sought, informational support given, and informational support sought, respectively.

| Social Support | Pearson's Correlation |
|---|---|
| Emotional Support Given | 0.400*** |
| Emotional Support Sought | 0.468*** |
| Informational Support Given | 0.503*** |
| Informational Support Sought | 0.547*** |

Table 2: Results from the analysis to determine the correlation between the social support models and the annotator ratings:. p-value<0.001: ***; p-value<0.01: **; p-value<: 0.05*.

## 4 Changes in Social Support Sought Over Time

In this section, we aim to determine how social support expressed in posts in our dataset changes over time. For posts published in the same week, we averaged their scores for emotional support given, emotional support sought, informational support given, and informational support sought, respectively. As shown in Figure 1, we observed that emotional support sought increased over time and the emotional support given increased in the beginning of March and reduced by April. Also, we observed that informational support given and sought followed a similar pattern, however, users published more posts in which they gave informational support, as shown in Figure 2.

## 5 Discussion and Future Work

This work has two main findings. In posts published in a 2-months time period in a COVID-19 online forum: (i) over time, users sought more emotional support and gave less emotional support and (ii) users gave more informational support compared to informational support sought.

These changes in social support needs can provide insights as to the well-being and needs of the public, thereby informing the forum moderators/members and the necessary authorities on the kinds of interventions to provide. For example, given that overtime, users in the forum sought

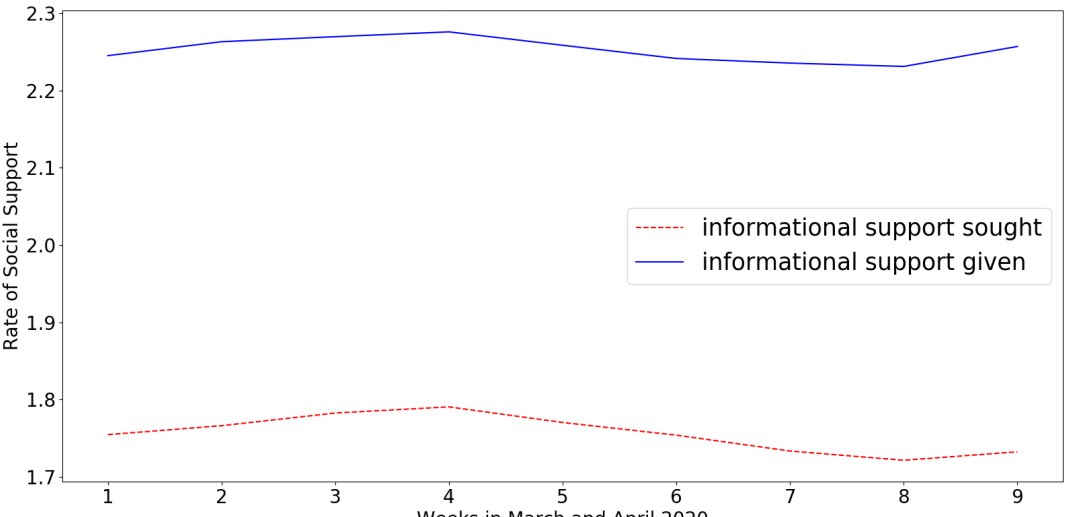

Figure 2: Graph showing the change in the average weekly informational support sought and informational support given in posts from our /r/Coronavirus dataset

| Examples |
|---|
| 1. Yesterday, Medicare expanded access to telehealth services, meaning your provider will get compensated from Medicare for you to do the appointment over the smartphone, computer, etc. |
| 2. There are now 430 thousand total cases of the virus around the world as of 9:35 eastern time. In the United States, the total cases are close to 55 thousand. Number of deaths worldwide is 19 and a half thousand, and the number of deaths in the United States are 781. |
| 3. Some good news today; rate of increase of new cases in New York appears to be slowing. An analysis of CNN's count shows that average rate of day-over-day increase for the last seven days was 17% - compared to 58% for the previous seven-day period. |

Table 3: Examples of posts giving informational support

more emotional support, online psychiatry services may be provided. Another finding from this work is that users in the forum tended to give more informational support. We observed that a significant number of posts were relaying information reported by or obtained from other sources such as news articles or websites, as shown in Table 3, which shows three of the top rated informational support given posts. In some cases, the veracity of the information given are not determined, hence the potential for the spread of misinformation about the pandemic; this highlights the importance of having experts engage with these forums in order to provide and verify information.

In Stokes et al. (2020), posts from an online forum focused on COVID-19 were analyzed and it was determined that longitudinal topic modeling of posts published in the forum can identify changes in public discussions around COVID-19. In the future, we aim to determine how the social support expressed around specific topics such as "COVID-19 symptoms" and "mental health" change over time in an online COVID-19 forum. In this work, we tracked social support changes in a 2-months time period; in the future, we aim to track these changes over a longer time period. The interests of members of different online forums differ, despite the similarity of the topics discussed on the forums (Tran and Ostendorf, 2016), hence in the future, we aim to analyze other online COVID-19 forums to determine the changes in social support expressed over time.

## 6 Conclusion

In this work, we build machine learning models to measure the extent of social support expressed in posts on an online forum focused on discussions around COVID-19 and determine how these social supports change over a period of 2-months. Since individuals are turning to online COVID-19 related forums to seek support, it is important to understand what type of social supports people are seeking on these forums and how they change over time.

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
