# OpenReview forum: "Understanding the Changes in Social Support Expressed in a COVID-19 Online Forum "
_EMNLP/2020/Workshop/NLP-COVID — Submitted to NLP-COVID19-EMNLP_

### Official Review · AnonReviewer1 · 2020-09-18
**Needs refinement in the analysis and statistical testing**

**Rating:** 4
**Confidence:** 4

**Review:**

Interesting paper and with addressing the comments below this could be a candidate for this workshop.

Comments:

- line: 46: define social support
- line 100: provide full details (of how the support types are distributed across your sample) in addition to the two examples
- related: consider adding examples for each type being low/high.
- line 155: great to see proper agreement evaluations!
- line 195: I'd really like to see comparisons with other models (e.g. SVM, simple linear regr., LASSO, etc.) - also: please add detail on the model used (did you treat the Likert scores as "classes" or predict continuous values?)
- line 200: the x-axis of the plot could do with refinement (e.g. actual week dates)
- line 227: the problem (predicting Likert-scale scores) can be formulated as a regression problem, so in addition to the r reported, I'd like to see regression performance metrics (at least: R-squared, RMSE and MAE).
- Fig 1: please assess statistically whether the lines in the plot are different - right now the interpretation relies on eye-balling.
- line 287-386 (Discussion section): you are now not talking about the prediction any more but focus on the findings derived from the plots. Please add a discussion of the prediction analysis and provide more rigorous testing of the plots.
- a key (and very interesting) finding is that the rate of support given/received is divergent for emotional and informational support - this merits more attention and I'd like to see more details on this (e.g. is it driven by specific posts /outliers; what is a possible explanation for that divergence).
Lastly: the interpretation of the RF findingfs (Table 2) deserve some review: r=0.40 means that the model merely explains 16% of the variance in the outcome variable (here: degree of support) - since R-squared = 0.16. This is not a lot so there will be many other factors aside from features used - I'd like to see this discussed in the paper.

---

### Official Review · AnonReviewer3 · 2020-09-21
**Interesting topic, unclear methodology**

**Rating:** 4
**Confidence:** 3

**Review:**

This paper addresses the topic of COVID-19 online support forums with the purpose of automatically classifying posts that either seek or receive emotional or informational support. While the topic of the paper is of interest for this workshop's audience, the paper could be improved both in terms of presentation of the methodology and results (as well as their discussion).

The authors exclude comments to the original posts during the collection of the dataset. I did not understand this decision, could the authors detail their reasoning? Aren't comments more prone to hold some kind of support utterance? If one user asks for a specific information I would expect that the comments associated to the original question to contain some kind of informational support. Section 3.1 should be discussed in more detail. I understand this is a short paper, however in the current form it wouldn't be possible to reproduce the work done by the authors. Which LIWC categories were considered reevant to emotional and informational support? How were the posts involving request or advice identified? What were the 20 topics generated using LDA? The results in Table 2 deserve a more thorough discussion.

---

### Official Review · AnonReviewer2 · 2020-09-24
**Interesting work, but methodology and presentation need to be improved**

**Rating:** 4
**Confidence:** 4

**Review:**

The study converted the covid19 suggestion seek/give problem into a social media text classification task, the data is from Reddit and the experiments showed some average weekly emotional support sought/given changes. I found the topic is interesting, but the presentation needs to be improved. it would be better if the authors can give more statistics of the collected dateset, show the  agreement level of the annotators, given some reasons of the feature selection,  illustrate basic parameter settings for the model Random Forest (e.g. how many trees and splits are there), and give the model performance level. In general , I think the work can still be improved.